# Formin 2 links neuropsychiatric phenotypes at young age to an increased risk for dementia

Roberto Carlos Agís-Balboa[1,‡] (iD), Paulo S Pinheiro[2,3,5,†,§,#], Nelson Rebola[2,3,†,§,#], Cemil Kerimoglu[1,4], Eva Benito[1], Michael Gertig[1], Sanaz Bahari-Javan[4], Gaurav Jain[1], Susanne Burkhardt[1], Ivana Delalle[5], Alexander Jatzko[6], Markus Dettenhofer[7] (iD), Patricia A Zunszain[8], Andrea Schmitt[9,10], Peter Falkai[9], Julius C Pape[11], Elisabeth B Binder[11], Christophe Mulle[2,3], Andre Fischer[1,4,*] (iD) & Farahnaz Sananbenesi[1,12,**] (iD)

## Abstract

Age-associated memory decline is due to variable combinations of genetic and environmental risk factors. How these risk factors interact to drive disease onset is currently unknown. Here we begin to elucidate the mechanisms by which post-traumatic stress disorder (PTSD) at a young age contributes to an increased risk to develop dementia at old age. We show that the actin nucleator Formin 2 (*Fmn2*) is deregulated in PTSD and in Alzheimer's disease (AD) patients. Young mice lacking the *Fmn2* gene exhibit PTSD-like phenotypes and corresponding impairments of synaptic plasticity, while the consolidation of new memories is unaffected. However, *Fmn2* mutant mice develop accelerated age-associated memory decline that is further increased in the presence of additional risk factors and is mechanistically linked to a loss of transcriptional homeostasis. In conclusion, our data present a new approach to explore the connection between AD risk factors across life span and provide mechanistic insight to the processes by which neuropsychiatric diseases at a young age affect the risk for developing dementia.

**Keywords** aging; Alzheimer's disease; Formin 2; HDAC inhibitor; post-traumatic stress disorder
**Subject Categories** Molecular Biology of Disease; Neuroscience
The EMBO Journal (2017) 36: 2815–2828

See also: **J Gräff** (October 2017)

## Introduction

Alzheimer's disease (AD) is the most common neurodegenerative disorder causing a huge emotional and economical burden to our societies. The vast majority of AD cases are sporadic and arise on the background of variable genetic and environmental risk factors. There is substantial evidence that pathological changes occur years before the onset of clinical symptoms (Bateman *et al*, 2012). To elucidate how the different risk factors contribute to disease onset is therefore of utmost importance. The starting point of this study is epidemiological data indicating that individuals suffering from neuropsychiatric diseases such post-traumatic stress disorder (PTSD) at a young age have an increased risk of developing AD in old age (Yaffe *et al*, 2010; Burri *et al*, 2013; Weiner *et al*, 2013; Stilling *et al*, 2014a). PTSD develops in response to a traumatic event when normal psychological defense mechanisms fail. Individuals that suffer from PTSD are deficient in learning that a stimulus associated with an adverse event no longer presents a threat, a process that requires cognitive flexibility and extinction learning (Goswami *et al*, 2013). Memory extinction can be analyzed in rodents using the well-established fear conditioning paradigm. This paradigm represents a specific form of learning that underlies the reduction of previously acquired fear memories (Myers & Davis,

1 Department for Epigenetics and Systems Medicine in Neurodegenerative Diseases, German Center for Neurodegenerative Diseases (DZNE) Göttingen, Göttingen, Germany
2 Interdisciplinary Institute for Neuroscience, University of Bordeaux, Bordeaux, France
3 CNRS UMR 5297, Bordeaux, France
4 Department of Psychiatry and Psychotherapy, University Medical Center Göttingen, Göttingen, Germany
5 Department of Pathology and Laboratory Medicine, Boston University School of Medicine, Boston, MA, USA
6 Department of Psychosomatics, Westpfalzklinikum-Kaiserslautern, Teaching Hospital, University of Mainz, Mainz, Germany
7 CEITEC – Central European Institute of Technology, Masaryk University, Brno, Czech Republic
8 Stress, Psychiatry and Immunology Laboratory, Institute of Psychiatry, Psychology & Neuroscience, King's College London, London, UK
9 Department of Psychiatry and Psychotherapy, LMU Munich, Munich, Germany
10 Laboratory of Neuroscience (LIM27), Institute of Psychiatry, University of Sao Paulo, São Paulo, Brazil
11 Department of Translational Research in Psychiatry, Max Planck Institute of Psychiatry, Munich, Germany
12 Research Group for Genome Dynamics in Brain Diseases, Göttingen, Germany
 *Corresponding author. Tel: +49 551 3961211; E-mail: andre.fischer@dzne.de
 **Corresponding author. Tel: +49 551 39 61211; Fax: +49 3961212; E-mail: farahnaz.sananbenesi@dzne.de
 † These authors contributed equally to this work
 ‡ Present address: Psychiatric Diseases Research Group, Galicia Sur Health Research Institute, Sergas, Cibersam, Complexo Hospitalario Universitario de Vigo (CHUVI), Vigo, Spain
 § Present address: CNC-Center for Neuroscience and Cell Biology, University of Coimbra, Coimbra, Portugal
 # Corrections added on 2 October 2017 after first online publication: The author name has been corrected and present address added.

2002; Lattal *et al*, 2006; Sananbenesi *et al*, 2007; Fischer & Tsai, 2008; Radulovic & Tronson, 2010). Although impaired fear extinction in rodents does not recapitulate the complex phenotypes observed in humans suffering from PTSD, it is suitable to study the mechanisms that underlie increased susceptibility to PTSD. Thus, we reasoned that a promising strategy to elucidate the molecular mechanisms that link PTSD at a young age to an increased risk for age-associated memory decline and AD would be to screen for animal models that exhibit (i) impaired fear extinction at young age, while consolidation of new memories is still intact, but (ii) develop accelerated memory impairment in the presence of additional AD risk factors. By this, we identified the *Formin 2* (*Fmn2*) gene. FMN2 is best known for its role in regulating actin dynamics (Schuh, 2011) and was previously detected in a screen for genes that are deregulated in the aging mouse hippocampus (Peleg *et al*, 2010). In addition, FMN2 has been linked to synapse formation and deletion mutations of the *Fmn2* gene are associated with intellectual disability, pointing to a role for *Fmn2* in memory function in mice and humans (Peleg *et al*, 2010; Almuqbil *et al*, 2013; Law *et al*, 2014). Here we show that loss of *Fmn2* affects plasticity at the mossy fiber-CA3 synapse and causes impaired fear extinction in young mice. We furthermore show that *Fmn2* expression is decreased in PTSD patients and in post-mortem brain samples from AD patients. Loss of *Fmn2* accelerates age-associated memory decline, which is further accelerated in the presence of amyloid pathology and is accompanied by deregulation of hippocampal gene expression. While the mechanisms that link reduced FMN2 levels to aberrant gene expression are likely to be multifactorial, we provide evidence that FMN2-dependent synaptic actin dynamics signal via the ERK1/2 pathway to drive Elk1 and SP1-dependent gene expression. Of note, memory impairment in all employed models is ameliorated after oral administration of the HDAC inhibitor Vorinostat. In sum, our data represent a new approach to explore the cross-talk between AD risk factors across life span and provide evidence that loss of transcriptional plasticity is a key event down-stream of AD risk factor exposure.

## Results

### Loss of FMN2 leads to impaired fear extinction in young mice

We started our analysis by employing animal models for age-associated memory decline and tested whether extinction phenotypes precede the impairment of memory consolidation. We decided to test APPPS1-21 and 5xFAD mice, two well-established mouse models for amyloid deposition and age-associated memory impairment that are linked to familiar Alzheimer's disease (Radde *et al*, 2006; Ohno *et al*, 2007; Govindarajan *et al*, 2013). In addition, we employed *Fmn2* knockout (*Fmn2*$^{-/-}$) mice. Although the role of FMN2 in the adult brain is not well understood, FMN2 has recently been linked to age-related memory impairment in mice (Peleg *et al*, 2010) and is associated with cognitive dysfunction in humans (Almuqbil *et al*, 2013; Law *et al*, 2014). Mice from all experimental groups were subjected to contextual fear conditioning training at 3 month of age. Freezing behavior, indicative of associative memory consolidation, was similar in all mutant mice when compared to the corresponding control groups during a memory test performed 24 h

later (Fig 1A). To test extinction of fear memory, animals were re-exposed to the conditioning context on consecutive days in the absence of the foot shock, a procedure that induces extinction learning and leads to the reduction in the aversive freezing response (Sananbenesi *et al*, 2007; Tronson *et al*, 2012). Fear extinction was similar in APPPS1-21 (Fig 1B) and 5xFAD mice (Fig 1C) when compared to the corresponding control groups. In contrast, fear extinction learning was impaired in *Fmn2*$^{-/-}$ mice (Fig 1D). Next we confirmed that all three lines of mutant mice exhibit impaired consolidation of new memories already at 8 months of age (Fig 1E), while wild-type mice were impaired only at 16 months of age (Fig 1F). These data indicate that *Fmn2*$^{-/-}$ mice develop deficits in fear extinction that precede impairment of memory consolidation. However, the possibility remained that the employed fear conditioning protocol does not allow for the detection of very mild memory impairments. Therefore, we also subjected *Fmn2*$^{-/-}$ mice to a milder fear conditioning protocol (Kerimoglu *et al*, 2013). In line with our initial observation, *Fmn2*$^{-/-}$ mice did not exhibit memory impairment at 3 months of age (see Appendix Fig S1). In line with these data, 3-month-old *Fmn2*$^{-/-}$ mice were able to acquire spatial reference memory in the Morris water maze test similar to wild-type mice (Fig EV1A and B). Reversal learning was, however, impaired (Fig EV1C and D), which supports the view that loss of *Fmn2* in young mice does not affect the consolidation of new memories but is required for changing responses to existing memories. Taking together, these data support the view that the analysis of FMN2 would be a good starting point to investigate the mechanisms by which neuropsychiatric diseases at young age contribute to an increased risk for age-associative memory decline. Before continuing our analysis, however, we sought to provide further evidence for this hypothesis. To this end, we tested *Fmn2* expression in blood samples from PTSD patients via qPCR. We observed a significant reduction in *Fmn2* expression in PTSD patients when compared to age-matched control individuals (Zieker *et al*, 2007; Fig 1G). Although care has to be taken when interpreting data obtained from blood in the context of brain diseases, there is increasing evidence that adverse life events can induce long-lasting changes in the expression of specific genes and that such changes in gene expression are detected in various cell types such as cells obtained from blood or saliva (Smith *et al*, 2015). Thus, the analysis of gene expression, for example, in blood, is viewed as a suitable approach to identify biomarker and surrogate marker for brain diseases (Rao *et al*, 2013; Ciobanu *et al*, 2016; Schmitt *et al*, 2016). The fact that *Fmn2* levels are altered in blood samples from PTSD patients therefore indicates that exposure to PTSD-inducing events may similarly altered *Fmn2* levels in the brain. Since we do not have access to suitable post-mortem tissue from PTSD patients, this hypothesis remains to be tested. However, PTSD and other neuropsychiatric diseases have been linked to altered glucocorticoid signaling (Du & Pang, 2015; Kim *et al*, 2015). It is therefore interesting that *Fmn2* expression was decreased in human neuronal progenitor cells subjected to acute or chronic dexamethasone treatment, a synthetic glucocorticoid that can have detrimental effects on neuronal function and cognitive abilities (Crochemore *et al*, 2005; Tongjaroenbuangam *et al*, 2013; Feng *et al*, 2015; Lanshakov *et al*, 2016; Fig 1H). We also observed decreased *Fmn2* expression in post-mortem human brain samples (hippocampus) from AD patients when compared to age-matched control individuals (Fig 1I).

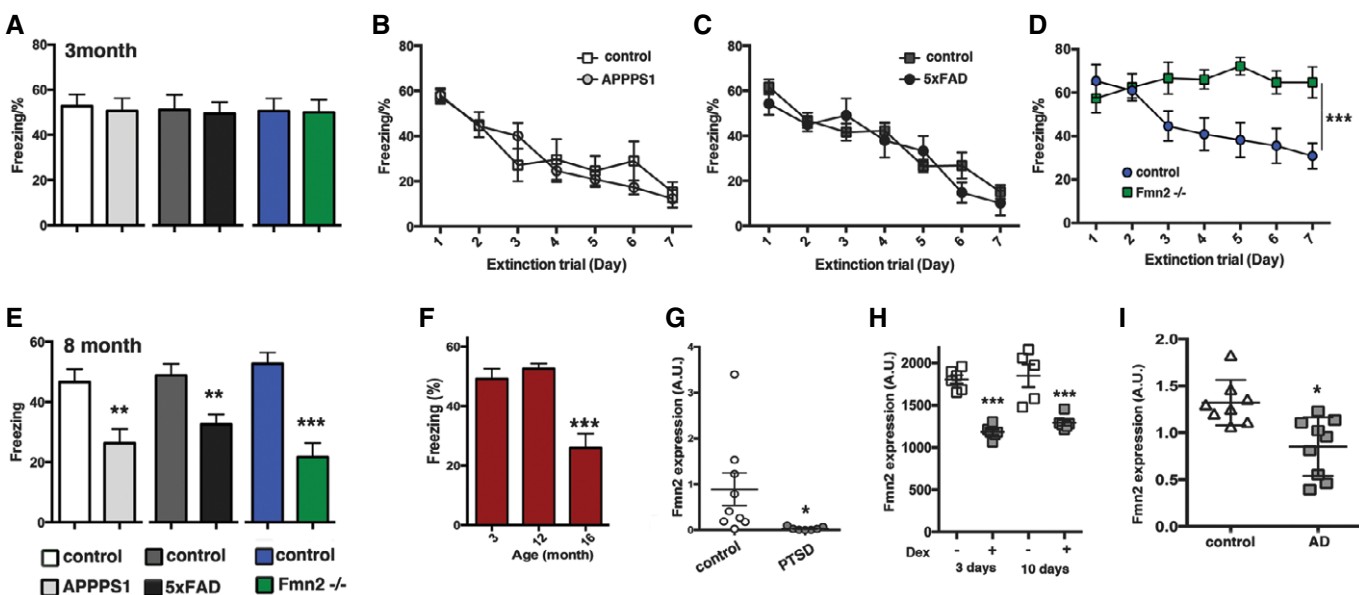

**Figure 1.   Impaired fear extinction precedes memory decline in *Fmn2* mutant mice.**

A   Freezing behavior tested 24 h after training was similar in 3-month-old APPPS1-21, 5xFAD, and *Fmn2*⁻/⁻ mice when compared to corresponding control groups (n = 10/group).

B   Fear extinction in 3-month-old APPPS1 mice is similar to a non-transgenic control group (n = 9/group).

C   Fear extinction in 3-month-old 5xFAD mice is similar to a non-transgenic control group (n = 9/group).

D   Extinction learning in the contextual fear conditioning paradigm is impaired in *Fmn2*⁻/⁻ mice (n = 9/group; ***P < 0.001, F = 21.77, two-way RM ANOVA).

E   Freezing behavior was tested 24 h after training in 8-month-old APPPS1-21, 5xFAD, and *Fmn2*⁻/⁻ mice. A significant impairment in associative memory was observed when comparing APPPS1 mice (n = 10) to a corresponding control group (n = 8; **P < 0.01, t-test), when comparing 5xFAD mice (n = 10) to a corresponding control group (n = 8; **P = 0.01, t-test), and when comparing *Fmn2*⁻/⁻ mice (n = 10) to corresponding *Fmn2*⁺/⁺ wild-type mice (n = 10; ***P < 0.0001, t-test).

F   Freezing was analyzed in 3- (n = 11), 12- (n = 9), and 16 (n = 8)-month-old male wild-type mice. A significant impairment (***P < 0.0001, t-test) was observed in 16-month-old animals when compared to either 3- or 12-month-old animals.

G   *Fmn2* expression measured in blood samples via qPCR was reduced in PTSD patients (n = 9) when compared to age-matched controls (n = 7; *P = 0.0361, t-test).

H   *Fmn2* mRNA expression is decreased after dexamethasone treatments in human hippocampal progenitor cells (HPCs) during neuronal proliferation (3 days) and differentiation (10 days) (***P_adj < 3.27 × 10⁻⁶, one-way ANOVA followed by Bonferroni correction).

I   qPCR revealed that Fmn2 expression is decreased (*P = 0.037, t-test) in AD patients (n = 9) when compared to age-matched controls (n = 8).

Data information: Error bars indicated SEM.

Encouraged by these observations, we decided to study the role of FMN2 in the adult brain in greater detail. *Fmn2* was highly expressed in mouse and human hippocampal neurons (Fig 2A). In line with these findings, robust FMN2 protein levels were observed in the mouse and human hippocampus (Fig 2B). The specificity of the FMN2 antibody employed for immunoblotting was confirmed using corresponding hippocampal tissue from *Fmn2*⁻/⁻ mice as negative control (Fig 2C). Since we could not identify a suitable antibody to reliably detect FMN2 via immunostaining, we generated *Fmn2*-EGFP knock-in mice. These mice are healthy and show no obvious phenotype (see Appendix Fig S1B and C). The analysis of FMN2 protein localization in FMN2-EGFP knock-in mice confirmed high levels of FMN2 protein in the hippocampus and especially in the mossy fiber pathway (Fig 2D), suggesting that FMN2 can localize to the pre-synaptic compartment. In line with this, we detected robust levels of FMN2 in hippocampal synaptosomes (Fig 2E). Furthermore, the pre-synaptic marker protein synaptoporin co-localized with FMN2 within the stratum lucidum of the hippocampal CA3 region (Fig 2E).

These data suggest that the susceptibility to develop PTSD-like phenotypes observed in 3-month-old *Fmn2*⁻/⁻ mice involves compromised plasticity at the hippocampal mossy fiber-CA3

synapse. Thus, we analyzed the electrophysiological properties of mossy fiber synapses in *Fmn2*⁻/⁻ mice at 3 months of age. We found that long-term potentiation (LTP) and long-term depression (LTD) were unaffected (Fig 2F and G). However, *Fmn2*⁻/⁻ mice exhibited impaired depotentiation, a well-established phenomenon by which mossy fiber-CA3 LTP can be reversed using long trains of low-frequency stimulation that has been discussed as a molecular correlate of extinction processes (Hong *et al*, 2009; Kim *et al*, 2009; Fig 2H).

Although gross brain morphology, motor coordination, explorative behavior, and basal anxiety were similar when comparing 3-month-old *Fmn2*⁻/⁻ mice and control littermates (Fig EV2), our *Fmn2*⁻/⁻ mice constitutively lack FMN2. Thus, the possibility remained that the observed fear extinction phenotype could in part be due to subtle development abnormalities and in addition may not be specific to hippocampal function. To test for this possibility, we employed an RNAi approach and found that RNAi-mediated decrease in FMN2 in the hippocampus impairs fear extinction, but has no effect on the consolidation of new memories. To this end, we first confirmed that intra-hippocampal injection of an siRNA against *Fmn2* reduces *Fmn2* mRNA and protein levels when compared to the scramble control group

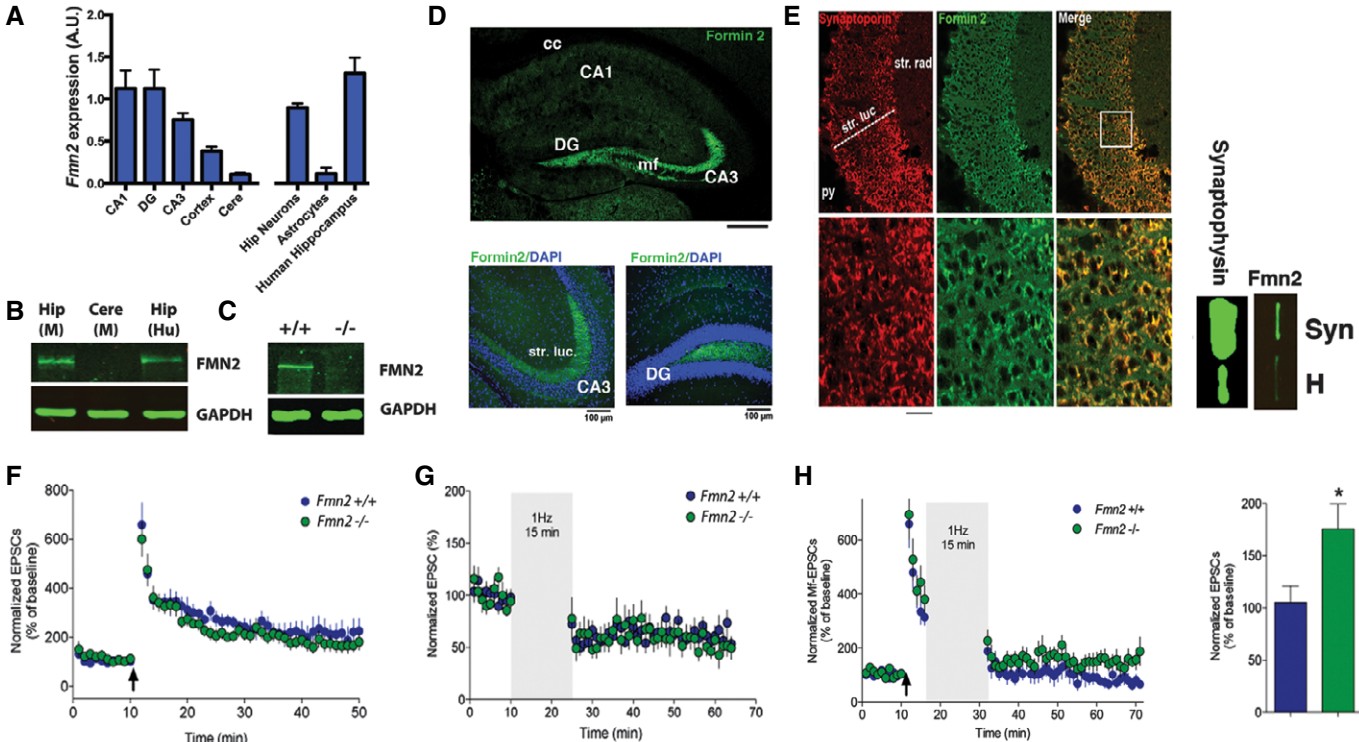

**Figure 2. FMN2 is highly expressed in mossy fibers and is required for synaptic depotentiation.**

A   *Fmn2* expression was analyzed via qPCR in brain regions from 3-month-old mice (left panel, *n* = 5) and in mouse primary hippocampal neurons, astrocytes (*n* = 5) and human post-mortem hippocampus (right panel; *n* = 3/group).

B   Representative immunoblot images showing FMN2 protein levels in the mouse hippocampus (Hip M), cerebellum (Cere M), and human hippocampus (Hip Hu).

C   Representative immunoblot images showing FMN2 levels in the hippocampus of wild-type and *Fmn2⁻/⁻* mice.

D   Representative low (upper panel, scale bar 500 μm) and high magnification images (lower panels, scale bars 100 μm) showing FMN2 expression (green) in FMN-EGFP knock-in mice.

E   Left panel: Representative images showing the co-localization of FMN2 (green) and synaptoporin in the mossy fiber pathway in low and high magnification. Right panel: Immunoblot analysis of hippocampal synaptosomes (Syn) compared to cell homogenate (H) showing that FMN2 is present in the pre-synaptic compartment. Synaptophysin was used to confirm enrichment of synaptosomes. Scale bar: 20 μm.

F   LTP was similar in *Fmn2⁻/⁻* (*n* = 12) and wild-type control mice (*n* = 8).

G   The absolute level of LTD was similar amongst *Fmn2⁻/⁻* and wild-type mice (*n* = 10).

H   Depotentiation was significantly impaired in *Fmn2⁻/⁻* mice (*n* = 12) when compared to control (*n* = 11; *P < 0.05, *t*-test).

Data information: For all experiments, parasagittal brain slices from 14- to 21-day-old mice were used. Cere, cerebellum; Hip, hippocampus, mf; mossy fiber, str. luc.; stratum lucidum, cc, corpus callosum. Error bars indicate SEM.

(Fig 3A–C). Next we subjected mice to contextual fear conditioning and injected siRNA or scrambled control RNA for 3 days (every 12 h) before animals were subjected to fear extinction (Fig 3D). Injections were continued during extinction trials 1-3. Extinction was significantly impaired when comparing the siRNA versus the scramble control RNA-injected mice (Fig 3E). When animals were allowed to rest for 6 days after extinction trial 4 and did not receive further injections, normal extinction behavior was observed when mice were subsequently subjected to further extinction training. These data indicate that siRNA-mediated impairment of extinction is not due to unspecific effects of the injection procedure (Fig 3D and E). Finally, we show that siRNA-mediated knockdown of *Fmn2* does not affect the acquisition of contextual fear memories (Fig 3F).

In conclusion, these data indicate that loss of FMN2 leads to impaired fear extinction which is accompanied by mild deficits in neuronal plasticity that mimic the behavioral alterations.

**Loss of FMN2 leads to accelerated age-associated memory impairment**

Next, we decided to explore the mechanisms by which FMN2-mediated phenotypes at a young age increase risk for late life dementia. We first confirmed our observation that 8-month-old *Fmn2⁻/⁻* mice exhibit impaired memory formation (see Fig 1E) in an additional memory test, namely the hippocampus-dependent Morris water maze paradigm. There was no difference in the escape latency when comparing 3- versus 8-month *Fmn2⁺/⁺* mice (Fig 4A). However, 8-month-old *Fmn2⁻/⁻* mice were significantly impaired when compared to age-matched wild-type mice (Fig 4B). This finding was confirmed in a probe test performed after 10 days of training. While 3- and 8-month-old wild-type mice showed a significant preference for the target quadrant, 8-month-old *Fmn2⁻/⁻* mice were significantly impaired (Fig 4B). These data confirm that loss of *Fmn2* accelerates age-associative memory impairment. Next

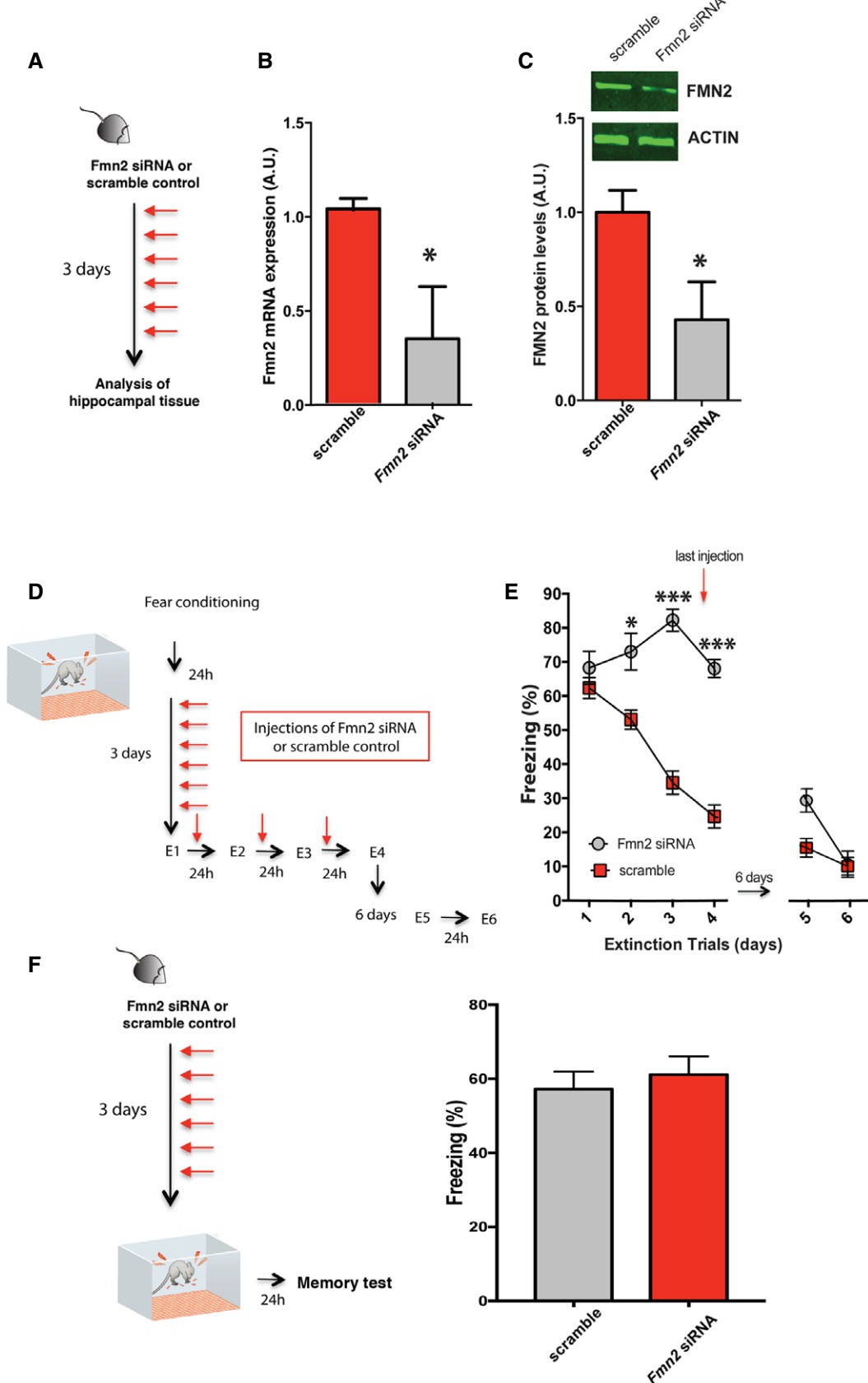

**Figure 3.**

we decided to investigate the impact of FMN2 on cognitive decline in the presence of another risk factor for AD, namely amyloid deposition. To this end, we employed APPPS1-21 mice. We observed that 3-month-old APPPS1-21 mice subjected to the open-field test and the contextual fear conditioning paradigm showed explorative (Fig 4C) and freezing behavior that was similar to wild-type mice, suggesting that at 3 months of age APPPS1-21 mice show no defect in associative memory formation (Fig 4D). Thus, we reasoned that the analysis in 3-month-old mice would enable us to detect synergistic effects on memory impairment in $Fmn2^{-/-}$ and APPPS1-21 mice. To this end, we crossed $Fmn2^{-/-}$ mice with APPPS1-21 (APP) mice and tested memory function in animals at 3 months of age. As expected, brain weight was not affected (see Appendix Fig S2). Moreover, explorative behavior in the open-field test did not differ significantly amongst groups (Fig 4C). When subjected to the fear conditioning paradigm, neither 3-month-old $Fmn2^{-/-}$ nor APPPS1-21 mice showed impaired freezing behavior in comparison with control littermates (Fig 4D). Freezing behavior was, however, significantly impaired in 3-month-old $Fmn2^{-/-}$_APPPS-21 mice (Fig 4D). Next we assayed spatial memory in the Morris water maze paradigm. All groups were able to acquire spatial memory throughout the 8 days of training (Fig 4E). When subjected to the probe test, all groups except $Fmn2^{-/-}$_APPPS1-21 mice showed a significant preference for the target quadrant (Fig 4F). In conclusion, these data show that reduced levels of FMN2 further accelerate memory decline in a mouse model for aging and in a model for amyloid deposition.

## Loss of FMN2 accelerates age- and amyloid-induced deregulation of gene expression

We have previously hypothesized that the various AD risk factors eventually cause aberrant gene expression, thereby contributing to the loss of homeostasis and memory decline (Sananbenesi & Fischer, 2009; Fischer, 2014). To test whether this hypothesis could help to explain how loss of $Fmn2$ at young age contributes to age-associative memory decline, we employed the hippocampal DG region for RNA-sequencing. First we compared gene expression levels in 3-month-old $Fmn2^{-/-}$ mice and age-matched control littermates. In addition to the $Fmn2$ gene, we detected only 26 differentially expressed sequences (Fig 5A; Dataset EV1).

Nevertheless, GO-term and functional pathway analysis identified "oxidative phosphorylation"—and especially subunits of the NADH dehydrogenase and ATPase—to be increased (Fig 5B and C). When we compared the gene expression in cognitively impaired 8-month-old $Fmn2^{-/-}$ mice to age-matched control littermates, we detected 461 differentially expressed genes, of which the majority were down-regulated (Fig 5A). GO-term and pathway analysis revealed that the genes deregulated were linked to ribosome function, RNA splicing, Alzheimer's disease, and oxidative phosphorylation (Fig 5D and F). While genes linked to oxidative phosphorylation were increased in 3-month-old mice, genes of the same complexes and especially genes encoding subunits of the NADH dehydrogenase, ATPase, and Cytochrome C reductase were all down-regulated in 8-month-old mice (Fig 5E, Dataset EV1). These data suggest that the susceptibility to develop PTSD-like phenotypes observed in 3-month-old $Fmn2^{-/-}$ mice is not accompanied by massive changes in gene expression, while memory impairment in 8-month-old $Fmn2^{-/-}$ mice correlates with substantial deregulation of transcriptome plasticity. The fact that the spliceosome appeared to be deregulated in 8-month-old $Fmn2^{-/-}$ mice is in line with recent data suggesting that during aging, synaptic plasticity genes are deregulated at the level of differential exon usage (Stilling *et al*, 2014b; Benito *et al*, 2015). Thus, we also analyzed differential exon usage. In 3-month-old $Fmn2^{-/-}$ mice, we detected 46 genes that showed altered differential exon usage when compared to 3-month-old control littermates (Fig 5G, Dataset EV1). No significant pathways were identified. The same comparison was performed in 8-month-old $Fmn2^{-/-}$ and age-matched control mice and revealed 286 differentially expressed exons (Fig 5G). Further GO-term and pathways analysis showed that processes affected by differential splicing in 8-month-old $Fmn2^{-/-}$ mice are linked to synapse function (Fig 5H). In line with the gene expression data, we also observed that processes linked to energy metabolism and oxidative phosphorylation were deregulated (Fig 5H). In conclusion, these data indicate that the PTSD-like phenotypes observed in $Fmn2^{-/-}$ mice are not linked to substantial changes in gene expression, while aberrant gene expression is accelerated in the context of aging as an additional AD risk factor. To further substantiate this finding, we also analyzed gene expression in the DG of 3-month-old $Fmn2^{-/-}$_APPPS-21. Comparing the gene expression in 3-month-old APPPS1-21 and age-matched wild-type control mice

---

◀ **Figure 3. Hippocampal FMN2 is essential for extinction learning and cognitive flexibility.**

A   Experimental design. Anti-*Fmn2* or scrambled siRNA oligomers were injected via microcannulae into the hippocampus of 3-month-old wild-type mice every 12 h for 3 days.

B   *Fmn2* mRNA levels were determined in the hippocampus of mice injected with either scrambled or *Fmn2* siRNA via qPCR (*n* = 4/group). *Fmn2* levels are reduced in mice injected with *Fmn2* siRNA when compared to the scrambled control group (*$P$ < 0.05, *t*-test).

C   Similarly, hippocampal FMN2 protein levels were decreased in mice injected with *Fmn2* siRNA when compared to the scramble control group (*$P$ < 0.05, *t*-test, *n* = 4/group).

D   Experimental design. Microcannulae were implanted into the hippocampus of wild-type mice. All mice were subsequently subjected to contextual fear conditioning training. *Fmn2* siRNA or scrambled control RNA was injected into the dentate gyrus 24 h later and then every 12 h for 3 days. On subsequent days, mice were subjected to extinction trial (E)1, E2, E3, and E4 (panel B). The injection protocol was continued until E3.

E   Fear extinction is significantly impaired ($P$ < 0.001, $F$ = 15.55; *$P$ < 0.05, ***$P$ < 0.001 for *post hoc* analysis) in 3-month-old mice that received intra-hippocampal injections of *Fmn2* siRNA when compared to the scrambled control group (*n* = 9/group). When siRNA injection was terminated, all mice exhibited normal extinction behavior.

F   Left panel: Experimental design. Right panel: Injection of Fmn2 siRNA before fear conditioning training does not affect the acquisition of fear memories (*n* = 10/group).

Data information: Error bars indicate SEM.

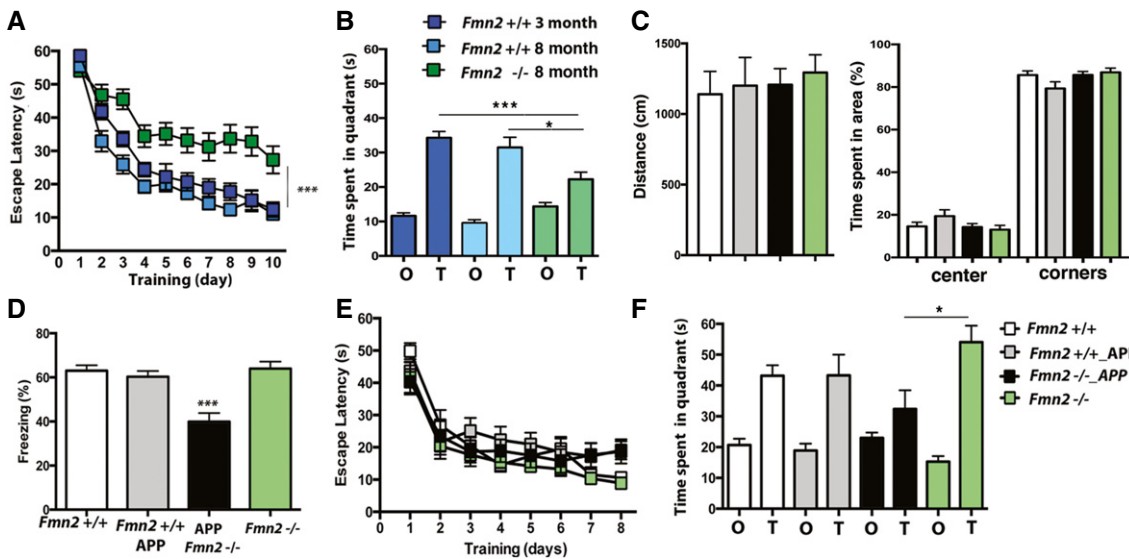

**Figure 4. FMN2 is linked to age-associated memory decline.**

A  Escape latency in 3-month-old *Fmn2^(+/+)^* mice (*n* = 12) compared to 8-month-old *Fmn2^(+/+)^* (*n* = 20) and *Fmn2^(−/−)^* mice (*n* = 20). Escape latency is significantly impaired in 8-month-old *Fmn2^(−/−)^* mice (***P* = 0.001, *F* = 24.1 versus age-matched *Fmn2^(+/+)^*, two-way RM ANOVA).

B  A probe test revealed that the time spent in the target quadrant significantly differed amongst groups (*P* = 0.0004, *F* = 9.4; one-way ANOVA) and that 8-month-old *Fmn2^(−/−)^* mice (*n* = 20) spent significantly reduced time in the target quadrant when compared directly to age-matched (**P* = 0.0143, *n* = 20, *post hoc* analysis) or 3-month-old *Fmn2^(+/+)^* mice (***P* < 0.0001, *n* = 12, *post hoc* analysis).

C  Left panel: The total distance traveled during a 5-min open-field exposure did not differ significantly amongst groups (*n* = 8/group). Right panel: The time spent in the center versus the corners during a 5-min open-field exposure was similar amongst groups (*n* = 8/group).

D  Contextual freezing in 3-month-old *Fmn2^(+/+)^* (*n* = 14), APPPS1-21 (*n* = 13), APPPS1-21_*Fmn2^(−/−)^* (*n* = 14), and *Fmn2^(−/−)^* mice (*n* = 14). Freezing behavior differed amongst groups (*P* = 0.0005, *F* = 6.947, one-way ANOVA) and was significantly impaired in APPPS1-21_*Fmn2^(−/−)^* mice when compared to the other groups (***P* < 0.0001, *post hoc* analysis).

E  Escape latency in the water maze test in 3-month-old *Fmn2^(+/+)^* (*n* = 15), APPPS1-21 (*n* = 15), APPPS1-21_*Fmn2^(−/−)^* (*n* = 15), and *Fmn2^(−/−)^* mice (*n* = 15). Comparing the learning curves amongst all groups revealed no significant difference.

F  During the probe test, there was a significant difference amongst groups in the time spent in the target quadrant (*P* = 0.04, *F* = 3.22, one-way ANOVA). Except for APPPS1-21_*Fmn2^(−/−)^* mice (*n* = 15), all groups showed a significant preference for the target quadrant (**P* < 0.05, *post hoc* analysis). A direct comparison between *Fmn2^(−/−)^* (*n* = 15) and APPPS1-21_*Fmn2^(−/−)^* mice (*n* = 15) revealed a significant reduction in the latter group (**P* = 0.017).

Data information: Error bars indicate SEM.

revealed 50 differentially expressed genes, while 268 differentially expressed genes were detected comparing 3-month-old APPPS1-21 mice to age-matched *Fmn2^(−/−)^*_APPPS1-21 mice (Fig 5I). The genes affected in 3-month-old APPPS1-21 mice were mainly linked to pathways representing inflammatory processes (Fig 5J). The top pathways affected in *Fmn2^(−/−)^*_APPPS1-21 mice when compared to APPPS1-21 mice were mainly linked to metabolic processes, suggesting that loss of Fmn2 does not accelerate Aβ-induced inflammation but is associated with deregulation of general cellular processes.

In sum, these findings show that loss of FMN2 in young mice leaves the hippocampal transcriptome rather unaffected. However, chronically reduced FMN2 levels accelerate deregulation of hippocampal transcriptome plasticity in response to aging or amyloid deposition.

Since our data indicate that FMN2 is a synaptic protein, we hypothesize that chronically low levels of FMN2 will eventually affect pathways linked to synapse-to-nucleus signaling (Jordan & Kreutz, 2009). FMN2 has been implicated with the regulation of actin dynamics, and it is interesting to note that changes in actin polymerization were found to signal to the nucleus and thereby induce gene expression changes (Olson & Nordheim, 2010). In

line with this, we observed that hippocampal actin dynamics were altered in *Fmn2^(−/−)^* mice (Fig EV3A). When we analyzed the promoter regions of genes deregulated in 8-month-old *Fmn2^(−/−)^* mice for the presence of any consensus sequences, we observed a significant enrichment for the motifs "CCCGCCCC" and "CCGGAAGC" which represents the ETS family of transcription factors (e.g., ELK) and the transcription factor specificity protein 1 (SP1), respectively (Fig EV3B). This is interesting, since SP1 and ELK have been linked to memory function and Alzheimer's disease (Sananbenesi *et al*, 2002; Sung *et al*, 2013; Szatmari *et al*, 2013; Citron *et al*, 2015; Wei *et al*, 2016) and are —amongst other kinases—activated via ERK1/2 (Bonello & Khachigian, 2004; Salim *et al*, 2007; Besnard *et al*, 2011; Kim *et al*, 2012). ERK1/2 is highly expressed in the mossy fiber pathway (Hu *et al*, 2004; Provenzano *et al*, 2014); has been linked to memory function, fear extinction, and Alzheimer's disease (Sweatt, 2004; Fischer *et al*, 2007; Kim & Choi, 2015); interacts with the actin cytoskeleton; and translocates to the nucleus upon stimulation (Wang & Hatton, 2007; Berti & Seger, 2017). It is thus tempting to hypothesize that chronically reduced FMN2 levels cause subtle change to the actin cytoskeleton that will eventually alter ERK1/2-SP1/ELK signaling leading to

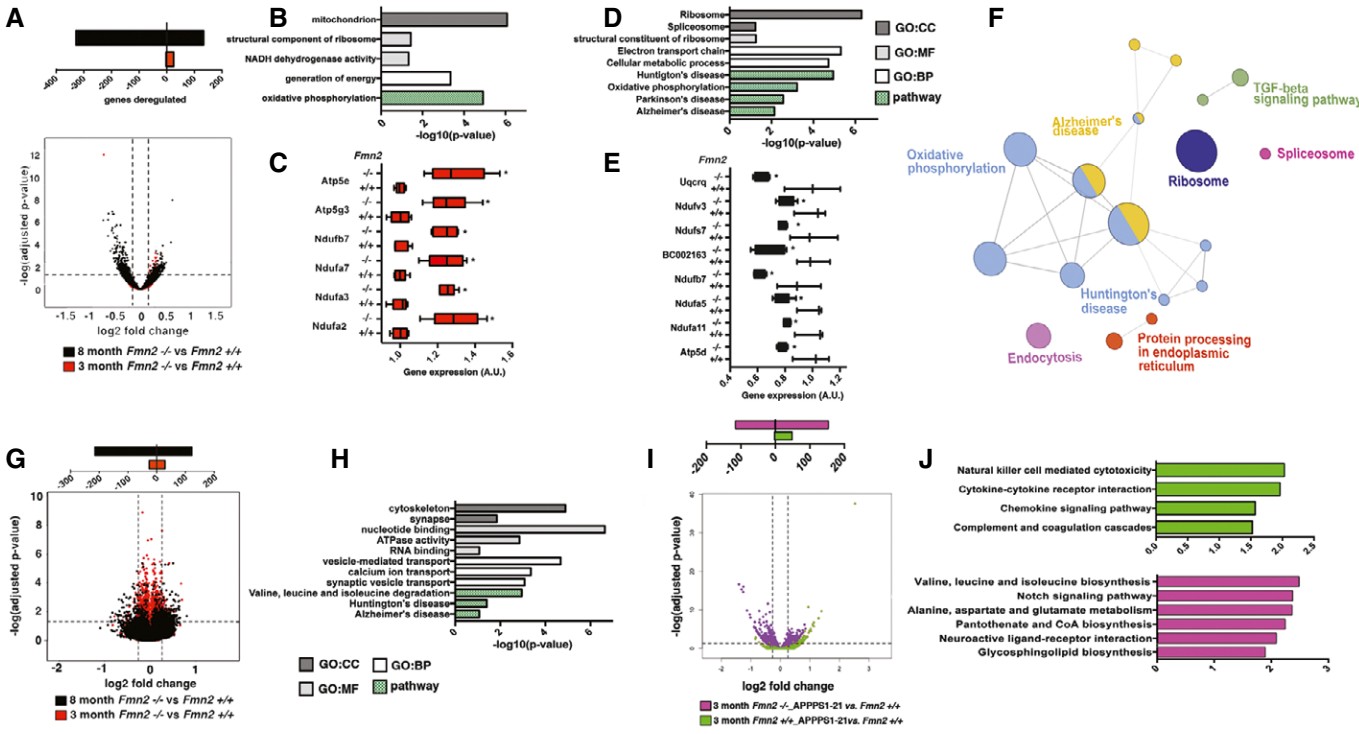

**Figure 5. Memory decline in 8-month-old *Fmn2*⁻/⁻ mice correlates with deregulated transcriptome plasticity.**

A Volcano plot showing genes differentially expressed in the dentate gyrus of either 3 (red)- or 8-month (black)-old *Fmn2*⁻/⁻ mice compared to corresponding wild-type control group (FDR < 0.05, log2 fold change < ± 0.25). The upper panel depicts the number of altered genes.

B Top GO domains and pathways affected in 3-month-old *Fmn2*⁻/⁻ mice when compared to wild-type control animals.

C Expression of selected genes affecting the oxidative phosphorylation pathway that are increased in 3-month-old *Fmn2*⁻/⁻ mice (*$P < 0.05$; *t*-test, $n = 5$/group).

D Top GO domains and pathways deregulated in 8-month-old *Fmn2*⁻/⁻ mice when compared to an age-matched control animals.

E Expression of selected genes of the oxidative phosphorylation pathway that were decreased in 8-month-old *Fmn2*⁻/⁻ mice (*$P < 0.05$; *t*-test, $n = 3$–5/group).

F Cytoscape-generated network of genes deregulated in 8-month-old *Fmn2*⁻/⁻ mice.

G Volcano plot showing differential exon usage in the dentate gyrus of either 3 (red)- or 8-month (black)-old *Fmn2*⁻/⁻ mice when compared to the corresponding wild-type control group (FDR < 0.05, log2 fold change < ± 0.2). The upper panel depicts the number of altered genes.

H Selected GO domains and pathways affected by differential exons usage in 8-month-old *Fmn2*⁻/⁻ mice when compared to a wild-type control group.

I Volcano plot showing genes differentially expressed in 3-month-old APPPS1-21 mice compared to a wild-type control group (green) and APPPS1-21_*Fmn2*⁻/⁻ mice compared to wild-type control mice (purple; FDR < 0.05, log2 fold change < ± 0.5). The upper panel depicts the number of altered genes.

J Functional pathways affected when comparing APPPS1-21 (green bars) or APPPS1-21_*Fmn2*⁻/⁻ mice (purple bars) to wild-type mice.

Data information: Error bars indicate SEM. GO:CC, gene-ontology domain "cellular compartment"; GO:MF, gene-ontology domain "molecular function"; GO:BP, gene-ontology domain "biological process"; pathway, "KEGG pathways".

aberrant gene expression. In line with this hypothesis, we observed that FMN2 and ERK1/2 interact and colocalize at the mossy fiber synapse (Fig EV3C–F) and that levels of active ERK1/2, SP1, and ELK1 are reduced in 8-month-old *Fmn2*⁻/⁻ mice (Fig EV3D). Although these data suggest one possible link between loss of FMN2 function and aberrant gene expression (Fig EV3G), certainly also other processes contribute to this phenotype. Irrespective of the multifactorial processes that link decreased FMN2 levels to aberrant gene expression, it is interesting to note that drugs aiming to state physiological gene expression such as inhibitors of histone-deacetylases (HDAC) were shown to improve learning behavior in mouse models for age-associated memory decline and amyloid deposition (Guan *et al*, 2009; Kilgore *et al*, 2010; Peleg *et al*, 2010; Govindarajan *et al*, 2011; Benito *et al*, 2015). Therefore, we wondered if HDAC inhibitors may also help to improve memory function in 8-month-old *Fmn2*⁻/⁻ mice and in 3-month-old *Fmn2*⁻/⁻_APPPS-21 mice.

### Vorinostat rescues memory impairment in aged *Fmn2*⁻/⁻ and *Fmn2*⁻/⁻_APP mice

Vorinostat (SAHA) is an FDA-approved HDAC inhibitor with beneficial effect on memory function (Kilgore *et al*, 2010; Benito *et al*, 2015). Thus, we decided to test its effect in aged *Fmn2*⁻/⁻ mice and in *Fmn2*⁻/⁻_APPPS1-21 mice. First, we treated 7-month-old *Fmn2*⁻/⁻ mice orally with Vorinostat (50 mg/kg) or placebo for 4 weeks. Explorative behavior measured in the open-field test was similar amongst groups (Fig 6A). Basal anxiety analyzed by comparing the center activity in the open field (Fig 6B) and by subjecting animals to the elevated plus maze test was also similar amongst groups (Fig 6C). Next we analyzed hippocampus-dependent associative memory in the contextual fear conditioning paradigm. Freezing behavior during the memory test was significantly increased in Vorinostat-treated *Fmn2*⁻/⁻ mice (Fig 6D). We also analyzed hippocampus-dependent spatial learning in the water

maze paradigm. When compared to the placebo group, $Fmn2^{-/-}$ mice treated with Vorinostat showed a significantly improved escape latency, indicative of improved spatial memory formation (Fig 6E). Swim speed was similar amongst groups (Fig 6E). During the probe test, only Vorinostat-treated $Fmn2^{-/-}$ mice showed a significant preference for the target quadrant (Fig 6F). We also tested the expression of selected genes linked to oxidative phosphorylation that were down-regulated in the DG of 8-month-old $Fmn2^{-/-}$ mice when compared to wild-type control littermates (see Fig 5). All selected genes were increased in the Vorinostat group when compared to placebo-treated $Fmn2^{-/-}$ mice (Fig 6G). HDAC inhibitors can affect gene expression by changing histone-acetylation. Reduced acetylation of histone 4 lysine 12 (H4K12ac) has been previously linked to age-associated memory decline (Guan et al, 2009; Peleg et al, 2010; Benito et al, 2015). In line with the gene expression data, we observed reduced H4K12ac in aged $Fmn2^{-/-}$ mice, which was, however, increased in mice treated with Vorinostat (Fig EV4). To further substantiate these findings, we performed a similar experiment in $Fmn2^{-/-}$_APPPS1-21 mice. Two-month-old $Fmn2^{-/-}$_APPPS1-21 mice were treated orally with either Vorinostat or placebo. Placebo-treated APP mice served as an additional control. Behavior in the open-field and elevated plus maze test did not differ amongst groups. However, similar to the findings in aged $Fmn2^{-/-}$ mice, we observed that Vorinostat treatment was able to ameliorate memory deficits, H4K12ac and gene expression in $Fmn2^{-/-}$_APPPS1-21 (Figs EV4 and EV5).

## Discussion

Our study was inspired by the observation that young individuals suffering from psychiatric diseases such as PTSD have an increased risk to develop AD as they age (Yaffe et al, 2010; Burri et al, 2013; Weiner et al, 2013). We reasoned that one possible way to begin elucidating this phenomenon would be to select genes that have been implicated with age-associative memory decline and to test whether these genes may also play a role in the development of PTSD-like phenotypes, which we analyzed in mice via fear extinction as a commonly used and robust paradigm. Nevertheless, we like to reiterate that results from animal models of neuropsychiatric diseases have to be interpreted with care, and while impaired fear extinction in rodents may point to the mechanisms that underlie increased susceptibility for PTSD, it does not fully recapitulate the phenotypes observed in PTSD patients. We observed that deficits in fear extinction precede memory decline in $Fmn2^{-/-}$ mice, and moreover, $Fmn2$ expression was decreased in PTSD and AD patients

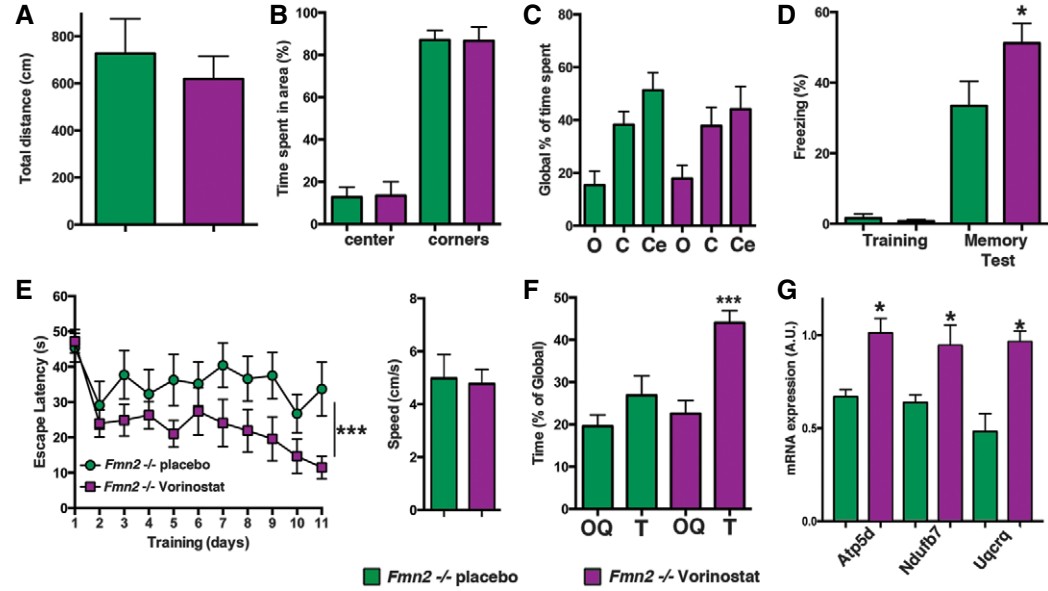

**Figure 6. Cognitive decline in 8-month-old $Fmn2^{-/-}$ mice is rescued by the HDAC inhibitor Vorinostat.**

A  The total distance traveled during a 5-min open-field exposure was similar amongst groups ($Fmn2^{-/-}$ placebo, $n$ = 9; $Fmn2^{-/-}$ Vorinostat, $n$ = 12).

B  The time spent in the center and the corners during a 5-min open-field exposure was similar amongst groups ($Fmn2^{-/-}$ placebo, $n$ = 9; $Fmn2^{-/-}$ Vorinostat, $n$ = 12).

C  The time spent in the open (o) and closed arms (c) or the center region (Ce) of an elevated plus maze was similar amongst groups ($Fmn2^{-/-}$ placebo, $n$ = 7; $Fmn2^{-/-}$ Vorinostat, $n$ = 10).

D  Freezing behavior during the memory test was significantly increased in the $Fmn2^{-/-}$ Vorinostat group when compared to the placebo group (*$P$ = 0.04, $t$-test; $Fmn2^{-/-}$ placebo, $n$ = 8; $Fmn2^{-/-}$ Vorinostat, $n$ = 9).

E  Left panel: The escape latency in the water maze test was significantly improved in the $Fmn2^{-/-}$ Vorinostat group when compared to the placebo group (***$P$ < 0.0001, $F$ = 4.86; two-way RM ANOVA, $n$ = 10/group). Right panel: Swim speed was not altered amongst groups.

F  Mice of the $Fmn2^{-/-}$ Vorinostat group spent significantly more time in the target quadrant (T) compared to the other quadrants (OQ). Please note the values for OQ are presented as average of all three other quadrants (***$P$ < 0.0001; *post hoc* analysis, $n$ = 10/group).

G  Expression of selected genes that were down-regulated in 8-month-old $Fmn2^{-/-}$ mice was significantly increased in the dentate gyrus from mice of the $Fmn2^{-/-}$ Vorinostat group when compared to placebo (*$P$ < 0.05; $t$-test, $n$ = 4/group).

Data information: Error bars indicate SEM.

and decided to follow up on this novel observation. We found that *Fmn2* is highly expressed in neurons of the mouse and human hippocampus and is especially enriched in the hippocampal mossy fiber pathway, where it is localized to pre-synaptic terminals. These data are in line with previous *in situ* hybridization findings showing that in the adult mouse brain *Fmn2* expression is highest in the hippocampal dentate gyrus (Schumacher *et al*, 2004). Nevertheless, it is likely that posttranslational mechanisms contribute to the enrichment of FMN2 at the mossy fibers. Interestingly, loss of *Fmn2* had no effect on LTP or LTD at the mossy fiber-CA3 synapse. Taking into account that LTP and LTD have been considered as molecular correlates of memory consolidation, these data are in agreement with the fact that young *Fmn2* mutant mice have no deficit in the consolidation of new memories. However, depotentiation was impaired in *Fmn2*$^{-/-}$ mice. It is tempting to link this observation to the behavioral phenotypes. Hence, at the synaptic and the behavioral level, mice that lack *Fmn2* at a young age are able to induce plasticity-related processes that are essential for information storage, but are impaired in modifying this information subsequently. These data are in line with previous studies, suggesting that depotentiation in the amygdala is required for fear extinction (Hong *et al*, 2009; Kim *et al*, 2009). The precise mechanisms by which loss of FMN2 affects depotentiation remain to be investigated. However, depotentiation has been linked to altered actin dynamics which in turn affects multiple synaptic processes such as vesicle trafficking (Galvez *et al*, 2016). We suggest that future research should test if such processes play a role in FMN2-mediated fear extinction. However, impaired depotentiation is certainly not the only process responsible for impaired fear extinction in *Fmn2*$^{-/-}$ mice.

A key observation of our study is that chronically reduced levels of *Fmn2* accelerate age-associative memory decline. The fact that in young *Fmn2*$^{-/-}$ mice the formation of hippocampus-dependent memories is not affected suggests that memory decline observed during aging is most likely an indirect consequence of the molecular changes triggered by chronically reduced FMN2 function. We hypothesize that such changes eventually lead to altered gene expression contributing to memory impairment. Indeed, we observed neglectable changes in hippocampal gene expression in 3-month-old *Fmn2*$^{-/-}$ mice, while loss of *Fmn2* dramatically increases the number of deregulated genes during aging or in response to amyloid pathology. These data support our idea that AD risk factors synergistically drive aberrant gene expression and thereby eventually contribute to dementia (Fig 7). Importantly, neither amyloid deposition nor FMN2 are likely to affect gene expression directly. While the precise mechanisms that couple FMN2-mediated defects in hippocampal function to deregulation of gene expression are likely multifactorial, it is interesting to note that stimuli which promote rearrangement of the actin cytoskeleton were found to induce gene expression changes (Olson & Nordheim, 2010). In line with this, we observe that loss of FMN2 affects actin dynamics which confirms previous data from other cellular systems (Pfender *et al*, 2011; Schuh, 2011; Montaville *et al*, 2016; Sahasrabudhe *et al*, 2016) and is in agreement with the fact that blocking actin dynamics in the hippocampus impairs fear extinction (Fischer *et al*, 2004). Alteration of actin dynamics has been associated with ERK1/2-mediated changes in gene expression (Wang & Hatton, 2007; Berti & Seger, 2017). Indeed, we found reduced ERK1/2 activity in *Fmn2*$^{-/-}$ mice. The finding that ERK1/2 activates the transcription factors ELK1 and SP1 is in line with our data showing that binding sites for these factors are enriched amongst the genes deregulated in aged *Fmn2*$^{-/-}$ mice (Fig EV3G). However, there are other processes linking synaptic plasticity to gene

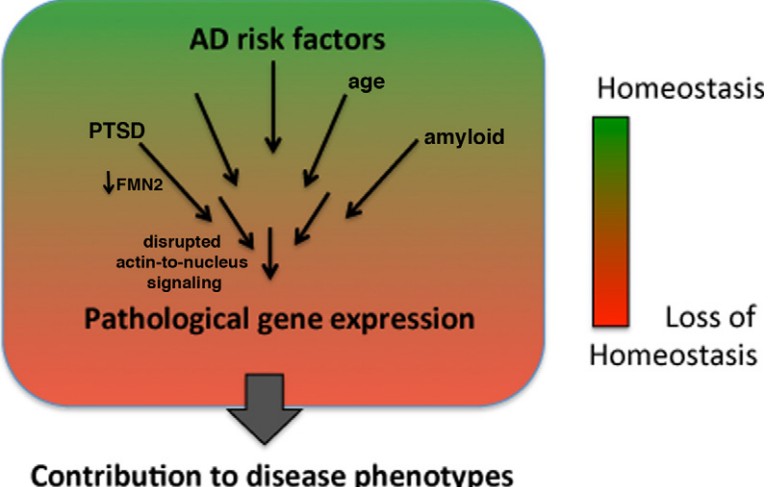

**Figure 7. AD risk factors cause loss of gene expression control.**

Our central hypothesis is that the various AD risk factors eventually lead to aberrant gene expression and loss of transcriptional control. This is based on the assumption that proper gene expression is a core feature of cellular homeostasis and that changes occurring at various compartments of the cell (e.g., at the synapse) will eventually signal to the nucleus and cause gene expression changes. In the case of FMN2, we provide, for example, evidence that chronically low levels of FMN2 disturb synaptic actin dynamics and thereby affect ERK1/2-dependent gene expression programs. Thus, even if an AD risk factor may play no direct role in gene expression, it may eventually contribute to a loss of transcriptional plasticity. We therefore suggest that targeting pathological gene expression could be a suitable therapeutic approach, especially for multifactorial diseases such as AD, where it is nearly impossible to determine all of the genetic and environmental factors that eventually contribute to clinical phenotypes.

expression (Jordan & Kreutz, 2009) that likely also contribute to a loss of transcriptional homeostasis. The view that disturbed transcriptome plasticity causatively contributes to memory loss is supported by our observation that the HDAC inhibitor Vorinostat reinstates hippocampal memory formation in aged $Fmn2^{-/-}$ mice and also in APPPS1-21 mice that lack $Fmn2$.

In conclusion, our data provide insight to the molecular mechanisms by which neuropsychiatric diseases at a young age lead to an increased risk for dementia when individuals age. We suggest that our experimental approach should be applied to additional genes and environmental factors implicated with PTSD or other neuropsychiatric diseases. Our data also provide a more general insight of how the various AD risk factors (in our case, FMN2-mediated PTSD-like phenotypes, amyloid deposition, and aging) contribute to dementia. Targeting transcriptome plasticity would thus be a very suitable therapeutic approach that is independent of the precise knowledge to the upstream pathological events (Fig 7). In line with this, we show that the HDAC inhibitor Vorinostat reinstates hippocampal memory formation in aged $Fmn2^{-/-}$ mice and also in APPPS1-21 mice that lack $Fmn2$. Of course, we cannot exclude that the memory enhancing effect of Vorinostat also involves processes not directly related to gene expression. It is, however, interesting to note that HDAC inhibitors are also discussed as a novel therapeutic avenue to treat PTSD (Whittle & Singewald, 2014). In fact, Vorinostat not only improves memory function, but also facilitates extinction of fear memories (Whittle *et al*, 2013), a finding we could confirm in our paradigm (see Appendix Fig S3). Our data therefore indicate that it might be possible to develop therapeutic strategies for PTSD patients that at the same time lower the risk to develop Alzheimer's disease, a line of research that should also be taken into consideration for other neuropsychiatric diseases.

# Materials and Methods

For detailed description of methods, see the Appendix Supplementary Methods.

### Animals and human tissue

Male mice were housed under standard conditions with free access to food and water. All experiments were carried out in accordance with the animal protection law and were approved by the District Government of Germany. $Fmn2^{-/-}$ and APPPS-21 mice have been described before (Leader *et al*, 2002; Radde *et al*, 2006). Three-, 12-, and 16-month-old wild-type mice were obtained from Janvier Labs. Post-mortem hippocampal tissue from AD patients and controls was obtained with ethical approval from the Alzheimer's disease Research Center Brain bank at Massachusetts General Hospital, Boson, MA, and from Brigham & Women's Hospital Autopsy Service, Boston, MA, USA. Samples were matched for age and post-mortem delay. AD patients Braak and Braak stage were 3–5. EDTA blood samples from PTSD patients and controls were from the Ramstein cohort described previously (Zieker *et al*, 2007). These individuals were exposed to the trauma of the air show catastrophe in Ramstein, Germany, 1988. We employed nine age- and gender-matched controls ($46.38 \pm 4.363$ years of age) and seven PTSD patients ($52.29 \pm 5.515$ years of age). Patients were diagnosed

according to DSM IV (SKID-I; Wittchen *et al*, 1997). Symptom severity was measured using the German versions of the clinician-administered PTSD scale [CAPS] (Schnyder & Moergeli, 2002; mean = 40.0, SD = 22.9; > 40 moderate PTSD), the Posttraumatic Stress Diagnostic Scale [PDS] (Foa *et al*, 1993; mean = 18.8, SD = 11.1; > 20 moderate to severe PTSD), the 22-item Impact of Event Scale Revised [IES-R] (Horowitz *et al*, 1979; mean = 48.0, SD = 18.7), and the Questionnaire on Dissociative Symptoms [FDS] (Freyberger *et al*, 1999; mean = 9.8, SD = 8.5; > 8.4 dissociative symptoms). To characterize possible comorbidity with a depressive disorder, the German version of the Beck Depression Inventory [BDI] (Hautzinger, 1991) was performed (mean = 14.3, SD = 9.1; > 18 relevant depression). None of the patients had been taking psychoactive drugs on a regular basis 3 months prior to the study. All experiments were approved by the local ethical committees, and tissue/blood samples were obtained upon informed consent.

### Behavior analysis

Mice were subjected to a series of behavioral tests as described before (Kerimoglu *et al*, 2013).

### Cannulation and siRNA injection

Intra-hippocampal injections were performed as described previously (Bahari-Javan *et al*, 2012).

### Statistical analysis

Unless specifically mentioned otherwise, data were analyzed by unpaired Student's *t*-test, two tailed *t*-test, Bonferroni test for multiple comparisons, or one and two-way and ANOVA (analysis of variance) when appropriate. Errors are displayed as standard error of mean (SEM). Unless otherwise stated, analysis was performed using GraphPad Prism.

### Gene expression

Analysis of RNA-sequencing has been described before (Stilling *et al*, 2014a). Data have been deposited to GEO database: GSE100070.

**Expanded View** for this article is available online.

## Acknowledgements
This work was supported by the following third party funds to AF: DFG research group KFO241/PsyCourse Fi981-4 and Fi981 11-1, DFG project 179/1-1/2013, an ERC consolidator grant (DEPICODE 648898), and the BMBF project Integrament; PF was supported by the KFO241/PsyCourse Project FA 241/16-1 (PF); FS was supported by the DFG grant SA1005/2-1. RCAB was supported by a Ramón y Cajal grant (RYC-2014-15246) and by Galicia Innovation Agency (IN607D-2016/003). We thank George Lu (University of Michigan) for proofreading the manuscript.

## Author contributions
RCA-B performed most of the experiments; PP, NR, and CM performed electrophysiological measurements; CK, EB, GJ, and SB performed gene expression analysis; MG and SB-J helped with behavioral experiments related to siRNA

injections; ID provided post-mortem human brain tissue; AJ, AS, and PF provided blood samples from PTSD patients; PAZ, JCP, and EBB performed experiments in human neuronal progenitor cells; AF and FS designed and coordinated the experiments and wrote the manuscript. MD generated FMN2-EGFP mice.

## Conflict of interest

The authors declare that they have no conflict of interest.

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
