## [Review Process File · The EMBO Journal]

Manuscript EMBO-2017-96821

Formin 2 links neuropsychiatric phenotypes at young age to an increased risk for dementia

Roberto Carlos Agis-Balboa, Paulo da Silva Pinheiro, Nelson Rebola, Cemil Kerimoglu, Eva Benito, Michael Gertig, Sanaz Bahari-Javan, Gaurav Jain, Susanne Burkhardt, Ivana Delalle, Alexander Jatzko, Markus Dettenhofer, Patricia A Zunszain, Andrea Schmitt, Peter Falkai, Julius C Pape, Elisabeth B Binder, Christophe Mulle, Andre Fischer, Farahnaz Sananbenesi

Corresponding author: Andre Fischer, DZNE

Review timeline:

Submission date:	25 February 2017
Editorial Decision:	07 April 2017
Revision received:	17 May 2017
Editorial Decision:	02 June 2017
Revision received:	23 June 2017
Accepted:	27 June 2017

Editor: Karin Dumstrei

Transaction Report:

1st Editorial Decision

07 April 2017

Thank you for submitting your manuscript to The EMBO Journal. Your study has now been seen by three referees and their comments are provided below.

As you can see, the referees find the analysis interesting and timely. However, they also find that further analysis is needed to substantiate the link between PTSD and neurodegeneration. The referees raise a number of constructive comments below that I would like to ask you to address in a revised version.

As you know, it is EMBO Journal policy to allow only a single round of revision and that it is therefore important to resolve the raised concerns at this stage. Let me know if we need to discuss anything in further detail

When preparing your letter of response to the referees' comments, please bear in mind that this will form part of the Review Process File, and will therefore be available online to the community. For more details on our Transparent Editorial Process, please visit our website: http://emboj.embopress.org/about#Transparent_Process

Thank you for the opportunity to consider your work for publication. I look forward to your revision.

REFEREE REPORTS

Referee #1:

Summary: This is an interesting and timely paper examining a role for Formin 2, a putative synaptic modulatory gene/protein in hippocampus, at the intersection of declarative memory that may be involved in both recovery from fear (extinction) in PTSD and deficits in fear memory in hippocampus. The authors perform a variety of elegant and interesting experiments in mouse and human, providing a fair amount of complementary, convergent data on the role of Fmn2 in memory formation. However, the paper is at times disjointed in its logic, jumping from one finding to the next without a connecting narrative at times, with some interpretations made without needed data to fully support such interpretations. Overall this is an important manuscript, but a number of concerns remain that limit enthusiasm at this time, as outlined below.

Concerns:

In results and figure EV2, morris water maze reversal learning is referred to as extinction learning, e.g. "Impaired extinction learning of 3 months old Fmn2 $-/-$ mice was also observed in the hippocampus-dependent Morris water maze paradigm (Fig. EV2)." While this may involve similar brain processes / structures, it is generally not referred to as extinction learning. I'd recommend limiting 'extinction' to decreased fear responding (or appetitive responding) to an aversive or appetitive cue, and using 'reversal' learning to describe changed responses to spatial / cognitive memory paradigms.

More discussion is needed related to the observation that Fmn2 is dysregulated both in blood (presumably mononuclear cells?) and brain. Are these similar or different mechanisms mediating these changes across tissues? This is particularly confusing given the synaptic role that the authors put forth for Fmn2 in the hippocampus.

The first sentence page 5, " This data suggest that hippocampal FMN2 is essential for fear extinction" seems misplaced, as this is a purely correlational finding at this point in the manuscript without causal data that hippocampal fmn2 is necessary or sufficient for fear extinction.

I think the RNAi knockdown of Fmn2 in hippocampus is a critical causal experiment, and would suggest placing it as a primary figure and not supplemental.

Are the authors speculating that FMN2 protein directly interacts with ELK protein at the synapse? If so, a co-IP type experiment of synaptosomal protein would be very helpful to more mechanistically support this hypothesis.

The connection between FMN2 and histone acetylation is weak and a bit confusing. What is the specificity of Vorinostat/SAHA? Which HDACs? Is it affecting hippocampal histone acetylation? Connection between histone regulation and Fmn2 expression?

Minor:

Figure 2D - typo - 'Fomrin'

'data' is often referred to as singular, whereas the more standard english usage is plural, such that 'These data' is more appropriate than 'this data'

Typo page 5, 1/3 way down "but has not effect on the consolidation of new memories" should read "but has no effect..."

On page 7, "substantialize" is not a word, i think you mean 'substantiate'

Referee #2:

This MS suggests that a single gene, Formin2, is involved in increased susceptibility to both PTSD at young age and AD at old age. Consistently, the authors showed that Formin2 is down-regulated in both PTSD and AD human patients. In a mouse model (Formin2 knockouts) at young age, the

absence of Formin2 produced the impairment in fear extinction and its cellular substrate (depotentialization at mossy fiber-CA3 synapses). The loss of Formin2 also induced AD phenotypes at old age. The authors suggested that the extinction impairment was due directly to the loss of Formin2 but the AD phenotype came from the altered transcriptional homeostasis after the loss of Formin2 was maintained for a long time. Finally, AD phenotypes in Formin2 knockouts was reversed by an epigenetic drug that is thought to enhance learning capability by changing gene expression profiles.

First of all, the authors should be applauded due to the thoroughness of their studies. Also, this MS provides one nice mechanism for extinction of contextual fear conditioning. In fact, there have been few studies for a cellular mechanism of extinction of contextual fear conditioning. In addition, it is very interesting to know that the loss of a single gene, Formin2, induces two different disease states sequentially, which could be a good example for the future studies. I only have several comments to improve this MS.

1. In the present MS, the impairment in fear extinction in mice was interpreted as the onset of PTSD. However, I am not sure whether this is the case. Rather, it could be interpreted as increased susceptibility to PTSD.

2. The authors need to clarify the meaning of the word, 'link' throughout their MS. Some readers might misunderstand that PTSD at young age somehow induces AD at old age.

3. There are many typos and errors throughout the MS (figure labeling, no years in references, typos etc.).

Referee #3:

In the article "Formin 2 links neuropsychiatric phenotypes at young age to an increased risk for dementia" Agis-Balbao and colleagues attempt to link PTSD-like symptoms to AD-related behavioral pathologies via formin 2 (*fmn2*). *Fmn2*^{-/-} mice display extinction deficits, which precede overall cognitive decline with age and there is some evidence that PTSD patients (in blood) have similarly reduced *fmn2* levels than AD patients (in the hippocampus). Upon crossing *fmn2*^{-/-} with APPPS1-21 mice, a well-established mouse model for AD, the authors then go on to show several pathways by whole genome and exon usage analyses that are affected. Finally, treatment with an established HDAC inhibitor, Vorinostat, seems to reverse some cognitive deficits in these mice.

This study touches upon an interesting link between PTSD and neurodegeneration, in particular AD, which has been alluded to in the past, but was never really shown experimentally. However, after reading the manuscript, I don't think that this link has been substantiated. Plus, there are some important experiments and control experiments missing, which dampen the enthusiasm for publication in the EMBO Journal.

Major points:

- No open field behavior of *fmn2*^{-/-} crossed to APPPS1-21 mice is presented.
- No control experiments/panels/explanations are shown for the shRNA experiments.
- The pathway analysis in the RNA-Seq experiments is somewhat meaningless. How can categories like ribosome, a cellular compartment, be compared to cellular metabolic process, a cellular function, to Alzheimer's disease, a pathology (all in panel D of figure 4)?
- In the HDAC inhibitor experiments, all types of behavioral tests are reported, but the most important one: Memory extinction. Which, according to the authors themselves (figure 1) is very important for PTSD.
- Also in the HDAC inhibitor experiments: What about the effect of Vorinostat on *fmn2*^{-/-} crossed to APPPS1-21 mice?

Minor points:

- Figure 2a, d: There is a clear mismatch between gene expression and protein expression data for formin2.
- Figure 2d: Formin2 is spelled wrong.

- What is the link of the mossy fiber synapse deficit to PTSD?
- Is there a link of aberrant gene expression patterns in the DG to PTSD?

1st Revision - authors' response

17 May 2017

Referee #1:**Referee 1, point 1:**

He/she says *"In results and figure EV2, morris water maze reversal learning is referred to as extinction learning, e.g. "Impaired extinction learning of 3 months old Fmn2 -/- mice was also observed in the hippocampus-dependent Morris water maze paradigm (Fig. EV2)." While this may involve similar brain processes / structures, it is generally not referred to as extinction learning. I'd recommend limiting 'extinction' to decreased fear responding (or appetitive responding) to an aversive or appetitive cue, and using 'reversal' learning to describe changed responses to spatial / cognitive memory paradigms."*

We really appreciate this insightful comment and changed the text accordingly. Please see **page 4, lines 12-16** of the revised manuscript.

Referee 1, point 2:

"More discussion is needed related to the observation that Fmn2 is dysregulated both in blood (presumably mononuclear cells?) and brain. Are these similar are different mechanisms mediating these changes across tissues? This is particularly confusing given the synaptic role that the authors put forth for Fmn2 in the hippocampus."

We apologize that this issue was not sufficiently explained. We address this point now in greater detail on **page 4, lines 23-32** of the revised manuscript. In brief, we were not trying to state that changes in *Fmn2* levels in blood contribute to PTSD-like phenotypes. Rather, there is increasing evidence that adverse life events can induce long-lasting changes in the expression of specific genes and that such changes in gene-expression can be detectable in various cell types such cells obtained from blood or saliva. Thus, the analysis of gene-expression, for example in blood, is viewed as a suitable approach to identify bio- and surrogate marker for brain diseases. The fact that *Fmn2* levels were altered in blood samples from PTSD patients therefore only indicate that exposure to PTSD can alter *Fmn2* and that similar changes may occur in the brain. This hypothesis could however not been tested since suitable post-mortem tissue from PTSD patients was not available to us.

Referee 1, point 3:

The first sentence page 5, " This data suggest that hippocampal FMN2 is essential for fear extinction" seems misplaced, as this is a purely correlational finding at this point in the manuscript without causal data that hippocampal fmn2 is necessary or sufficient for fear extinction.

We agree with this comment and have removed this sentence.

Referee 1, point 4:

I think the RNAi knockdown of Fmn2 in hippocampus is a critical causal experiment, and would suggest placing it as a primary figure and not supplemental.

We are very thankful for this encouraging comment. Due to space limitations we had placed this figure as "expanded view, EV". Importantly, we understood that EMBO J now offers the possibility to present data as EV figures that will be part of the main manuscript in the online version and thus will be immediately available to the readership without the need to search and download what was previously "supplemental material". Thus, for now, we decided to keep this data as Fig EV4 but we added more explanation of this data to the main text. Please see **page 5, lines 23-35** of the revised manuscript.

Referee 1, point 5:

Are the authors speculating that FMN2 protein directly interacts with ELK protein at the synapse? If so, a co-IP type experiment of synaptosomal protein would be very helpful to more mechanistically support this hypothesis.

This is a very interesting suggestion. As outlined in Fig. EV6 we suggest that FMN2 affects ERK1/2 signaling that eventually controls ELK/SP1-dependent gene-expression. As such, we do not propose that FMN2 and ELK would directly interact but rather suggest that FMN2 and ERK1/2 interact at the synapse and that ERK1/2 subsequently activates ELK in the cytoplasm. In support of this interpretation, ERK1/2 is known to localize to synapses but we are unaware of similar data related to ELK proteins. Indeed, even recent proteomic data sets on synaptosomes identify ERK, but not ELK proteins at synapses (e.g. see the data set published by Palmfeld et al., 2016, PMID: 27105822). To address this reviewer's question more specifically, we have now analyzed the levels of pERK1/2 and pELK-1 in hippocampal synaptosomes via immunoblot analysis. While pERK1/2 was detectable in synaptosomes, pELK-1 was absent. We also performed co-immunoprecipitation for pERK1/2. As expected we did not observe interaction with p-Elk1 in synaptosomes but could detect FMN2. These data confirm our immunohistochemical findings shown in panel C of Fig EV6. To further clarify the interpretation of our results we have now also included a model that summarizes how we envision the impact of FMN2 on gene-expression. See novel panels E-F of Fig EV6 of the revised manuscript. We also describe this novel data **on page 8, line 34, page 11, line 23, page 23, lines 24-37 and page 24, lines 1-4** of the revised manuscript.

Referee 1, point 6:

The connection between FMN2 and histone acetylation is weak and a bit confusing. What is the specificity of Vorinostat/SAHA? Which HDACs? Is it affecting hippocampal histone acetylation? Connection between histone regulation and Fmn2 expression?

We appreciate this question and apologize for any confusion. The rationale for testing if the HDAC inhibitor Vorinostat could reinstate memory function in *Fmn2*^{-/-} models was based on our finding that age-accelerated memory decline in *Fmn2*^{-/-} mice correlates with aberrant gene-expression. There is multiple data (including work from our group) suggesting that HDAC inhibitors can ameliorate age- and AD-associated changes in gene-expression (in mouse models) and thereby help to reinstate cognitive function (e.g. see Kilgore et al., 2010 PMID 20010553; Benito et al, 2015 PMID 26280576). Out of the available HDAC inhibitors we selected Vorinostat since it is approved for the use in humans and we are currently starting a clinical trial testing Vorinostat in AD patients (VorinostaAD01). The mechanisms by which HDAC inhibitors such as Vorinostat affect gene-expression include histone-acetylation but also non-histone acetylation-related processes. To specifically, address this reviewer's concern we provide novel data to strengthen the link between aberrant gene-expression and histone-acetylation in our experimental system. To this end we show that the selected genes decreased in aging *Fmn2*^{-/-} mice display reduced acetylation of histone 4 lysine 12 (H4K12), a histone-modification linked to age-associated memory decline (Peleg et al., 2010, PMID 20448184). Administration of Vorinostat rescues H4K12 acetylation in the promoter regions of these genes. Similar results were obtained for genes affected in *Fmn2*^{-/-} APP mice. This data is now shown as **novel FIG EV7** and described on **page 9, lines 28-32** and on **page 24, lines 7-22** of the revised manuscript.

Minor:

“Figure 2D - typo - 'Fomrin'“

We have now corrected this mistake.

“data' is often referred to as singular, whereas the more standard english usage is plural, such that 'These data' is more appropriate than 'this data'“

We have now carefully checked the use of the word “data” throughout the text of the revised manuscript.

Typo page 5, 1/3 way down "but has not effect on the consolidation of new memories" should read "but has no effect..."

We corrected this mistake.

On page 7, "substantialize" is not a word, i think you mean 'substantiate'
We corrected this mistake.

Referee #2:

Referee 2, point 1:

He/she says” *This MS suggests that a single gene, Formin2, is involved in increased susceptibility to both PTSD at young age and AD at old age. Consistently, the authors showed that Formin2 is down-regulated in both PTSD and AD human patients. In a mouse model (Formin2 knockouts) at young age, the absence of Formin2 produced the impairment in fear extinction and its cellular substrate (depotentialization at mossy fiber-CA3 synapses). The loss of Formin2 also induced AD phenotypes at old age. The authors suggested that the extinction impairment was due directly to the loss of Formin2 but the AD phenotype came from the altered transcriptional homeostasis after the loss of Formin2 was maintained for a long time. Finally, AD phenotypes in Formin2 knockouts was reversed by an epigenetic drug that is thought to enhance learning capability by changing gene expression profiles.*

First of all, the authors should be applauded due to the thoroughness of their studies. Also, this MS provides one nice mechanism for extinction of contextual fear conditioning. In fact, there have been few studies for a cellular mechanism of extinction of contextual fear conditioning. In addition, it is very interesting to know that the loss of a single gene, Formin2, induces two different disease states sequentially, which could be a good example for the future studies. I only have several comments to improve this MS.

1. In the present MS, the impairment in fear extinction in mice was interpreted as the onset of PTSD. However, I am not sure whether this is the case. Rather, it could be interpreted as increased susceptibility to PTSD. “

We are thankful for these encouraging results. We have now carefully checked the manuscript and re-wrote the corresponding passages to stress the fact that the analysis of fear extinction in mice is of course not recapitulating all complex phenotypes associated with PTSD in humans. Thus the mechanisms linked to impaired fear extinction in mice could indeed be viewed as processes that may increase susceptibility to PTSD. For example see **page 2 lines 34-36** from bottom, **page 5 line 36**, **page 7, line 21-22** or **page 10, lines 11-14**.

Referee 2, point 2:

He/she says “*2. The authors need to clarify the meaning of the word, 'link' throughout their MS. Some readers might misunderstand that PTSD at young age somehow induces AD at old age. “*

We are thankful for this insightful remark. We were not intending to propose that PTSD induces AD but that the mechanisms increasing the susceptibility to PTSD may also increase the risk to develop AD in the presence of additional AD risk factors. We have now changed the wording accordingly. For example see **page 2, line 34-37**, **page 4, line 18** of the revised manuscript.

Referee 2, point 3:

He/she says” *3. There are many typos and errors throughout the MS (figure labeling, no years in references, typos etc.).”*

We have now carefully checked the manuscript and to corrected all remaining mistakes-

Referee #3:

Referee 3, point 1:

He/she says “*No open field behavior of fmn2^{-/-} crossed to APPPS1-21 mice is presented.”*

We believe that this reviewer refers to the data shown in Fig 3. Indeed we did not show open field behavior in this set of experiments but we do show similar experiments as part of former Fig EV7 in

panel A. This figure now became Fig. EV8 in the revised manuscript. Here, panel A shows that there is no difference in the distance traveled in the open field when comparing placebo treated *Fmn2*^{-/-}_APP to APP mice and panel B shows that the center activity in the open field is not altered. We noticed, however, that the discussion of this data was limited in original manuscript. Thus, we now discuss this data in greater detail. Please see **page 9, lines 35-36** of the revised manuscript.

We also agree that it would be informative to show results of the open field test in mice that did not receive placebo treatment comparing *Fmn2*^{-/-}_APP directly to wild type mice. We had of course carefully tested the phenotype of the employed APP mice in pilot experiments in order to choose a time point for our experiments in which APP mice would not show a phenotype when compared to wild type mice. Thus we had also tested open field behavior. There was no difference amongst groups. As requested by this reviewer we have now included this data as novel panel C in Fig. 3 of the revised manuscript and describe these findings on **page 6, lines 23-24 & 29-30** and on **page 19, lines 27-30**.

Referee 3, point 2:

He/she says “*No control experiments/panels/explanations are shown for the shRNA experiments*”.

We believe that this comment is due to a misunderstanding. Especially since for example referee 1 specifically appreciated the siRNA experiment. Thus, we are confident that we have used proper control groups for siRNA treated mice, namely we injected scrambled RNA and we described the corresponding experiments/panels in the legend of Fig EV4 and in the methods section. We however appreciate this comment and have now added a more detailed description of the experiments to the main text. **Please see response to referee 1, point 4.**

Referee 3, point 3:

“*The pathway analysis in the RNA-Seq experiments is somewhat meaningless. How can categories like ribosome, a cellular compartment, be compared to cellular metabolic process, a cellular function, to Alzheimer's disease, a pathology (all in panel D of figure 4)?*”

We apologize for the confusion. The terms displayed in Fig 4D derived from the analysis of functional pathways and from the 3 domains of gene-ontology analysis, namely “cellular compartment”, “molecular function” and “biological process”. We agree that the presentation of the data could be improved and that it would be more informative to clearly indicate the corresponding analysis. Thus, we have now restructured the corresponding panels C, D and H of Fig 4. In panel J we only present pathways and we specify these issues now in the figure legend. Please see revised Fig. 4 and **page 20, lines 11,13-14, 21 & 25-27**.

Referee 3, point 4:

“*In the HDAC inhibitor experiments, all types of behavioral tests are reported, but the most important one: Memory extinction. Which, according to the authors themselves (figure 1) is very important for PTSD.*”

We believe that there is a misunderstanding. The rationale for testing if the HDAC inhibitor Vorinostat could reinstate memory function in *Fmn2*^{-/-} models was based on our finding that age-accelerated memory decline in *Fmn2*^{-/-} mice correlates with aberrant gene-expression. We discuss this issue now in greater detail and provide novel data to further strengthen the “Vorinostat data”. **Please see also response to “referee 1, point 6”.**

In addition we like reiterate that previous data have established that administration of Vorinostat facilitates fear extinction in mice. We discussed this data in the previous version of our manuscript (now page 12, lines 3-5). It is, however, true that the study by Whittle & Singewald employed a slightly different fear extinction protocol and a different HDAC inhibitor. To this end we have now tested the effect of Vorinostat (SAHA) administration in our paradigm. In line with the published data we observed that Vorinostat improves fear extinction in young mice. This data is now included as **novel Fig. EV10** and mentioned on **page 12, line 5-6** and **page 25, lines 17-23**.

Referee 3, point 4:

He/she says "Also in the HDAC inhibitor experiments: What about the effect of Vorinostat on *fmn2*^{-/-} crossed to *APPPS1-21* mice?"

We agree that this is an important question but we feel there is a misunderstanding. We had included this data in the original manuscript. The corresponding experiments had been described in Fig EV7, which now became Fig EV8 in the revised manuscript.

Minor points:

"Figure 2a, d: There is a clear mismatch between gene expression and protein expression data for *formin2*."

We agree that although *Fmn2* is expressed in all hippocampal subregions, the proteins appears to be highly enriched in the mossy fiber pathway. Although the underlying mechanisms remain to be investigated we speculate that posttranslational processes could contribute to this observation. We mention this issue now on **page 10, lines 21-22** of the revised manuscript.

"Figure 2d: *Formin2* is spelled wrong".

This mistake has been corrected.

"What is the link of the mossy fiber synapse deficit to PTSD?"

We had discussed in our manuscript that the "depotentialization phenotype" had been linked to fear extinction. Please see page 10, lines 26-31 of the revised manuscript. To the best of our knowledge it is, however, the first time that this link is reported for the mossy fiber pathway. To elucidate the precise mechanisms by which loss of synaptic *Fmn2* contributes to impaired depotentialization could thus be potential starting point to further explore the "*link of the mossy fiber synapse deficit to PTSD*". This view is supported by data linking depotentialization to actin dynamics, which affects multiple synaptic process such as for example vesicle trafficking. We are currently conducting further experiments related to this topic and hope to present data from this line of research within the next 3 years. We further discuss this issue now on **page 10, lines 31-35** of the revised manuscript.

"Is there a link of aberrant gene expression patterns in the DG to PTSD?"

In our experimental model PTSD-like phenotypes in *Fmn2*^{-/-} mice are not linked to massive changes in gene-expression (see Fig. 4A, 3 month group). However, it is important to note that we employ *Fmn2* knock out mice as our model system. The fact the *Fmn2* levels are changed in patients and in response to glucocorticoid signaling (See Fig 1 G-I) suggest that changes in gene-expression can contribute to the etiology of PTSD. This view is also in line with published data showing for example that subjecting mice to stressful stimuli that cause PTSD-like phenotypes, results in altered gene-expression in the amygdala (e.g. see Ponomarev et al., *Neuropsychopharmacology*, 2010; PMID: 20147889). Thus, the question asked by this reviewer is very interesting and insightful and should be addressed in future research using for example an experimental design similar to the one employed by Ponomarev et al. The outcome of such experiments will reveal additional candidate genes that could be analyzed in a similar manner as we describe it here for *Fmn2*. In fact, we suggested in the original manuscript that the approach used by us should be applied to additional genes and environmental factors implicated with neuropsychiatric diseases such as PTSD. Please see page 11, lines 30-32.

2nd Editorial Decision

02 June 2017

Thanks for submitting your revised manuscript to the EMBO Journal. Your study has now been re-reviewed by referees #2 and 3. Referee #1 was not available to review the revised version.

As you can see below, both referees appreciate the introduced revisions and support publication here. Referee #3 has a few remaining points that can be addressed with appropriate text changes.

REFEREE REPORTS

Referee #2:

All the concerns raised by the reviewers (1 & 2) have been adequately addressed.

Referee #3:

In the first round of revisions, I had raised the following major points:

- No open field behavior of *fmn2*^{-/-} crossed to APPPS1-21 mice is presented. The authors now provide these data and, importantly, a reference in the text to them. This addresses my concern.
- No control experiments/panels/explanations are shown for the shRNA experiments. The authors refer to EV 4 to address this point. I don't think that EV 4 addressed this issue, however. What was the scrambled sequence used in the shRNA experiment? What was the sequence of the siRNA? What are the coordinates? The authors cite one of their own study, but this is bothersome for the reader to figure out the details of the experiment. Did the authors actually hit the hippocampus during their injections? No injection sites are shown.
- The pathway analysis in the RNA-Seq experiments is somewhat meaningless. How can categories like ribosome, a cellular compartment, be compared to cellular metabolic process, a cellular function, to Alzheimer's disease, a pathology (all in panel D of figure 4)? The authors have now presented these data satisfactorily in the revised version of the manuscript.
- In the HDAC inhibitor experiments, all types of behavioral tests are reported, but the most important one: Memory extinction. Which, according to the authors themselves (figure 1) is very important for PTSD. The authors claim that it is not essential to test the effect of Vorinostat on memory extinction. I uphold my original criticism. *Fmn2*^{-/-} mice show clear deficits in extinction - an original finding by the authors. *Fmn2*^{-/-} mice show differential gene expression in the hippocampus, which can be - at least partially - rescued by vorinostat. To me, the next logical question remains the effect of vorinostat on memory extinction in these mice. If memory extinction impairments are rescued, this would be interesting. If they are not, then formin2 becomes even more interesting as a master regulator of extinction.
- Also in the HDAC inhibitor experiments: What about the effect of Vorinostat on *fmn2*^{-/-} crossed to APPPS1-21 mice? The authors had addressed this point already in the originally submitted manuscript. My oversight.

All minor comments have been addressed.

2nd Revision - authors' response

23 June 2017

Referee #3:

1.
He/she asks us to provide in the methods section the sequence of the *Fmn2* siRNA and the scrambled siRNA used in the experiments described and request to add more specifics regarding the injection procedure in Fig EV4.

Its true that in the main text we only refer to a previous study. However, we had provided detailed explanation of the experimental procedure in the supplemental information (now appendix; see "Cannulation and siRNA injection"). Regarding the sequences of the siRNAs we had stated that commercially available siRNA against *Fmn2* and scramble RNA were used and we provided the corresponding product information.

2.
He/she reiterates his/her question *about the effect of HDAC inhibitor on the effect of fear extinction*. The original comment read *"In the HDAC inhibitor experiments, all types of behavioral tests are reported, but the most important one: Memory extinction. Which, according to the authors themselves (figure 1) is very important for PTSD."*

We realize now that there has been a misunderstanding. In response to the original remark we had discussed previous data showing that HDAC inhibitors were found to affect fear extinction learning and in addition had provided novel data to show that Vorinostat facilitates fear extinction in wild type mice. It was not clear to us that the question actually referred to the effect of Vorinostat in *Fmn2*^{-/-} mice.

With respect to this question we like to state that we did not claim that *it is not essential to test the effect of Vorinostat on memory extinction*. We agree that this is a very interesting point in the context of *Fmn2*^{-/-} mice but as suggested by this reviewer it would be a *next logical question*. Thus, we believe that such experiments are beyond the scope of the present manuscript and are also not directly related to the message we communicate. In the present study we do not focus on the question if HDAC inhibitors can affect fear extinction but if Vorinostat can improve learning and memory in mice that had developed aberrant gene-expression due the combination of the AD risk factors “PTSD like phenotypes at young age”, “old age” and “amyloid pathology”. To test if Vorinostat would affect fear extinction in young *Fmn2*^{-/-} mice would in our view rather be the starting point of a new project. We agree with this reviewer that the outcome of such an experiment would in either case be very interesting and would allow us to further explore the molecular mechanisms underlying fear extinction. However, we like to reiterate that the outcome of the suggested experiment would rather be the starting point of a new project and we hope that we can present data related to this interesting suggestion in the future.

Corresponding Author Name: Andre Fischer

Journal Submitted to: EMBO J

Manuscript Number: EMBOJ-2017-96821